# Brazilian Urban Policy: Sustainability as a Driving Force

**Felipe Teixeira Dias** [1,2,3,*] , **Marcos Esdras Leite** [2] , **Priscila Cembranel** [3,4] , **José Baltazar S. O. de Andrade Guerra** [3] and **Robert S. Birch** [3,5]

1 Rede Colaborativa de Sustentabilidade Urbana & Direito à Cidade, Núcleo de Direito à Cidade no Semiárido, Observatório UniFG do Semiárido Nordestino, Guanambi 46430-000, BA, Brazil

2 Programa de Pós-Graduação em Desenvolvimento Social, Universidade Estadual de Montes Claros, Montes Claros 39401-089, MG, Brazil; marcos.leite@unimontes.br

3 Centre for Sustainable Development, University Southern of Santa Catarina, Florianopolis 88015-110, SC, Brazil; priscila.cembranel@unisociesc.com.br (P.C.); jose.baltazarguerra@animaeducacao.com.br (J.B.S.O.d.A.G.); r.s.birch@liverpool.ac.uk (R.S.B.)

4 Professional Master's Program in Administration, Department of Administration, Universidade do Contestado—UNC, Mafra 89306-076, SC, Brazil

5 School of Engineering, University of Liverpool, Liverpool L69 3GH, UK

* Correspondence: felipe.dias@sustentabilidadeurbana.com or felipeteixeiradias@gmail.com

**Definition:** Defining global themes such as Urban Policy, Urban Sustainability, and even the Right to the City (RTTC) is fundamental to stimulating and establishing a continuous dialogue with the scientific community, mainly in the social sciences. Thus, understanding the dynamics around the scope of urban sustainability requires an analysis that is focused on multiple global realities. Taking a holistic view of Brazilian Urban Policy, this entry looks at the historical contexts that make urban sustainability the driving force behind this policy. In addition, an interdisciplinary consideration of urban sustainability is proposed using an analysis that is based on the connection between urban policies and social functions that reflect the idea of a sustainable city. The results of this analysis also point to the need for a continuous debate on the subject that primarily promotes new discoveries; this is so that the driving force of urban policy can gain new meanings and new guidelines can be implemented.

**Keywords:** urban policy; urban sustainable development; right to the city; sustainable cities

## 1. Introduction

With the accelerated process of world urbanization [1], cities have become the focus of various studies and discussions [2], showing themselves to be fundamental to the discourse on social, economic and environmental development [3]. Cities have been considered using diverse approaches, judgements, advances and typologies that consider them as cultural, commercial, historical and religious centers in the development and adaption of contemporary models, i.e., to be sustainable and smart [4]. Thus, when examining the various social phenomena that permeate an urban network, it is necessary to understand and highlight the connection between urban policies and the meaning behind the slogan "Right to the City" (RTTC) [5]. In this context, the term "sustainable cities" has gained attention from various social sectors, both nationally or internationally [6]. As a result, it is important to highlight its relevance and emphasize the positive aspects of urban policies within the Brazilian Federal Constitution of 1988, which established the dynamics, mobilized the RTTC and, in turn, translated it into the idea that individuals have an inherent right to access urban sustainability [7]. Therefore, interlacing theoretical and practical discussions on urban policies, and their respective functionalities, reverberates strongly as a theoretical framework for developing an agenda of sustainable development [8].

The aim of this entry is to promote reflections on urban sustainability via an interdisciplinary approach whose objective is to interlink analyses connecting urban policies and

social functions that consider the idea of sustainable cities. This analysis of the nuances of urban policies for the promotion of sustainable cities is especially important when seeking answers from an array of theoretical assumptions and conjectures. In view of this ambiguity, the need for new studies and research on urban policy and sustainability is justified.

### 1.1. Delimiting Urban Policy

Contemplating, reflecting on and articulating contemporary themes that affect "urban space" has become almost inevitable insofar that urban spaces, rather than geographic spaces, have become the stage for various transformations [3] . Undoubtedly, among the various subjects that influence this discussion, dealing with urban sustainability under the aegis of urban policies [9] has become the theme in recent decades and will remain so for the foreseeable future [4]. In this way, the investigation of urban policies must be approached from two techno-scientific stances: an analysis of the founding principles and instruments employed for the execution of these so-called urban policies, and an analysis of the necessary geographic space that is linked to these formulated policies [6]. However, with regard to urban sustainability, there are several normative precepts that can direct the multiple analyses of this theme, i.e., the New Urban Agenda of UN-Habitat, or Objective number 11 of the United Nations 2030 Agenda [10]. Indeed, from the Brazilian perspective, this theme also gains strength via its incorporation into the City Statute Law 10.257/2001, which is legislation that proposes a series of fundamental guidelines and instruments that can be used to not only establish urban policies, but also to ensure urban sustainability [3].

Nowadays, there are several discussions, meanings and concepts held in the legal field about the RTTC. One such discussion of urban policies gained prominence in Brazil's constitution with the advent of the so-called Magna Carta of 1988 and, in doing so, strengthened its precepts and guidelines. From this perspective, urban policies have become strong instruments in assisting the processes inherent to the production, reproduction, and expansion of urban spaces in the Brazilian context [11]. Essentially, these aim to translate the constitutional determination into the orderly development of a city via its social functions, while guaranteeing the well-being of its inhabitants and their right to a sustainable environment [11]. Gradually, in Brazil, the process of creating and consolidating this legislation has been established, and it now embodies the Law of the City [12].

In this context, the "City Statute" has emerged as the main legislation; it aims to serve as the norm for the construction of ideas about a city and to simultaneously verify the processes that permeate the urban fabric [7]. From this perspective, the City Statute describes, both implicitly and explicitly, details behind the meanings of the RTTC and its compatibility with the precepts of urban policies [12]. Moreover, there are clashes and systematic distortions regarding the fundamental ideas and concepts that embody the idea related to urban policies and their alignment with the RTTC. Thus, such discussions on this theme are necessary in order to highlight these approaches and differences [13].

### 1.2. Model

For this entry, a qualitative approach is used to examine the relevant literature, draw conclusions and assemble a group of facts that are most pertinent to the past, present, and future contexts of urban sustainability. This strategy has the dual benefit of being both exploratory and descriptive, since the interconnection between urban policies and the functionalities of cities, for the advancement of urban sustainability, is explored and described in detail [14]. Finally, an integrative bibliographic research procedure that comprises the linking of bibliographies and conceptual assumptions is applied [15]. The main themes that emerge from this analysis are presented in Table 1, together with the principal references that cover them in the literature. Additionally, other authors that discuss and deepen the theories presented by the authors listed in Table 1 were used to supplement the database.

**Table 1.** Representation of the main authors and themes.

| Integrative Reference Portfolio | | |
|---|---|---|
| Theory | Authors | Themes/Title |
| Right to the City (RTTC) | [4,13] | Theory of RTTC |
| Urban Policy | [3] | Sustainable, Fair and Democratic Cities |
|  | [14,15] | What is a city? |
| Urban Sustainability | [16–18] | Right to urban sustainability |

## 2. RTTC and Brazilian Urban Policy

Since the whole terminology surrounding the construction of a "City" was coined in parallel with the idea of constructing a "State", it is noteworthy that the ancient Greeks called their community structures a "City-State" or polis. This idea ratifies the meaning that cities have autonomy within the territorial demarcation that is referred to as a State. In this sense, the State is a legal institution and is constituted by a larger legal document we call a "Constitution." Constitutions, as a maxim of norms that form, rule, and delineate a State, are thematically broad and promote a series of rights and guarantees of a fundamental nature.

In Brazil, the driving force dealing with the theme of urban policy gained support due to the advent of the promulgation of the Federal Constitution of 1988 and, subsequently, other city legislations have been woven around it. Thus, in becoming more normalized/enacted, this constitution acquired a fundamental status [9,18]. In giving potency to the issue of urban policy, the Statute of Cities Federal Law 10.257/2001, which regulates it, emerged as a historical milestone with respect to articles 182 and 183 of the Brazilian Magna Carta. This municipal law comprises several essential elements and promotes the full development of the social functions of a city.

Currently, there are several discussions regarding the existence or pursuit of new rights, and we should elucidate the conceptual distinction between the law of a city and the RTTC [19]; these are not synonymous, nor do they constitute a unit. Further discussion regarding the expression of RTTC evokes the concept espoused by Carvalho and Rodrigues [20], who reported that "at first, it is not a set of norms and principles that govern social relations in a city by delimiting their spatial organization, rules of conduct, functioning of institutions, mechanisms of popular participation, etc.". These aspects refer to urban law and, as a result, it is important to highlight that city law is a branch of urban law, since both connote the idea of the rules and principles that guide the urban policies. Clemente [12] reported that this comes from the "inherent foundations of the social function of cities that pursue the efficacy of city law".

Urban or city law aims to establish fundamental guidelines for the use and occupation of the land, to regulate the production of the urban space and to establish fundamental guidelines that are linked to and observed by the social functions of a city [7]. In this sense, highlighting that "the word law, contained in its definition, does not refer to the objective character of the term, but to the subjective, since it deals with the claim that the individual has to have access to a certain legal possession" [20]. In other words, the RTTC does not refer to the principled and normative body of rights that are specified by legislation, such as constitutional, civil or criminal law.

For a better understanding of the theme, Figure 1 illustrates the basic differences between the expressions of city law and RTTC. In summary, the propositions of the scholars Alfonsin, Clement, Lefebvre and Carvalho and Rodrigues [2,4,9,21], among others, refer to a claim, or even manifestation, in the search for the RTTC; this, in turn, refers to a whole set of rules and principles that aim to establish and promote a city that fulfils the social purposes of being more just, egalitarian and sustainable for its dwellers. In order to enable further understanding, the social functions of a city, which are correlated both to the

comprehension of the RTTC and to the guidelines of the urban policy, are examined more closely in the following section.

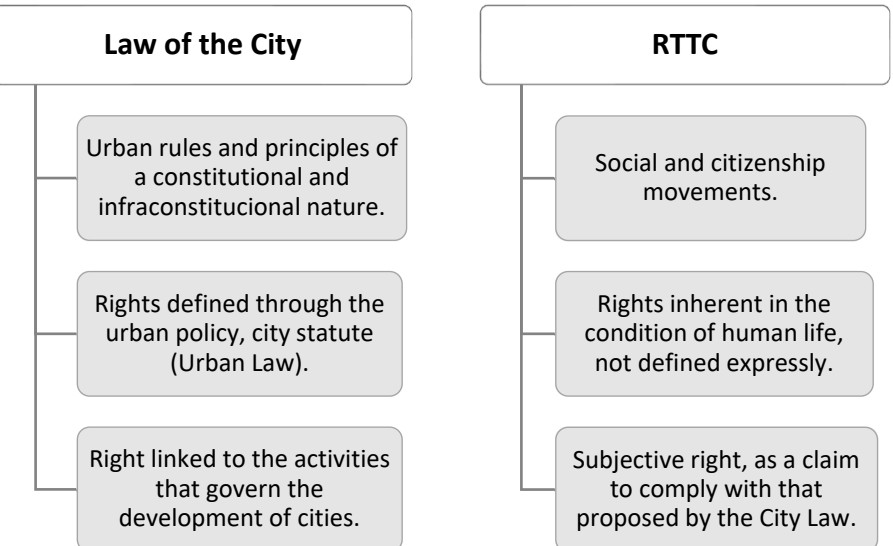

**Figure 1.** Basic distinctions between the Law of the City and the Right to the City (RTTC). Adapted from [4,9,21,22].

*Social Functions of a City as a Rule and as Principle*

The efficacy of city law is based on normative, legal and principled guidelines; in this sense, one of the guiding principles of the urban legal base is the social function of a city. This principle is also a guiding imperative of urban policy given the attention paid to it within the chapter dedicated to urban policies in the Brazilian Federal Citizen Constitution [6,8].

The conceptual views on this subject [23] are that, although the 1988 Constitution emphasizes the existence and need for urban legislation that fulfils the social functions of a city, it does not evidence what these functions would be. Therefore, under these circumstances, it is necessary to define and characterize these social functions.

Well-known authors, such as Lefebvre and Harvey [4,13], have already noted that the entire urban space, that is, cities and their functions, would favor social activities that simultaneously produce a collective good and give access to all city dwellers. As a result, Garcias and Bernardi [23] call attention to the issue by highlighting that the study of social functions is "permeated by sociological, philosophical, urbanistic, historical, economic and urban geographic concepts, among others." Therefore, the perception is that any study on the social functions of a city has an interdisciplinary and multidisciplinary nature, but does not lose the legal characteristics in the analysis of the norm. Following this line of thought, Clemente, Bernardi, Dias et al., and Maricato [9,16,19,22] point out that, above all, in order to have a right to a renewed and transformed urban life, it is necessary to pay attention to social functions. However, the following question remains: what are the social functions? However, evidently, the answer to this question is not to be found in the normative constitutional body.

Garcias and Bernardi [23,24] inferred that an interpretation or continuation of the understanding of the social functions of a city unfolds from the Athens Charters. However, as these social functions are not defined specifically, the door is open to new definitions of this principle. From this perspective, Clemente and Bernardi [9,19] elucidate that the social functions of the city are intricately linked to human dignity and the full development of urban activities, but that they also provide better living conditions. In the view of Lefebvre [5], being a product of the various modelling agents of urban space has several functionalities and these must meet the social concerns so that the RTTC can exist. In

general, Lefebvre, Carvalho and R. Rodrigues [4,21] describe social function as both a rule and a principle. As a rule, social function is provisioned in three spheres of law, Constitutional, Urban and Civil; meanwhile, questions about what they are and where they are admitted. As a principle, it is used as a categorical imperative beyond the legal normative body or is being evoked by the RTTC. From these discussions, Table 2 provides examples of the possible social functions of a city, but this is far from exhaustive.

**Table 2.** Classifications of social functions of a city.

| Urban Functions | Citizenship Functions | Management Functions |
|---|---|---|
| Housing | Education | Provision of Services |
| Work | Health | Urban planning |
| Leisure | Safety | Preservation of Cultural and Natural Heritage |
| Mobility | Protection | Urban Sustainability |

Adapted from: [19,20].

Given the above, it is understood that the social function of a city, besides being a rule, is also a guiding principle of urban policy. It is also noteworthy that, although it is not stated expressly how many social functions there are and what they are, this requirement may be embodied by new functions that arise from the sociability of the city. Logically, social functions are linked to the process of modifying urban space and the social needs of each epoch [18,24]. By corroborating with this thought, the work of Santos [25] highlights that no society has permanent functions as these change with the needs of each epoch by adapting to the reality and space. To this end, it is necessary to build a robust understanding from the knowledge of how a city is produced and how it relates to its producers [26].

### 3. Urban Sustainability as a Driving Force of Urban Policy

At the end of the nineteenth century, several researchers began to investigate the problem of urban chaos and environmental challenges, which began to integrate into the daily life of global realities [27]. This dynamic passed through several historical landmarks until a divergent perspective emerged in France in 1968 as a conception of the philosopher and sociologist Henri Lefebvre. It is the idea of the RTTC, a concept full of theoretical frameworks, that directs scientists to consider that urban and rural spaces must remain in harmony and connected [5]. It happens that advances in the RTTC have culminated erroneously in the preservation and unity of urban norms being disconnected from the natural environment and, thus, going on to cause new challenges [21].

In fact, in the Brazil of 1988, two achievements were enshrined by the Federal Constitution: the framework of the urban policy and the insertion of the environmental policy in the body of the Brazilian Magna Carta [11]. It was some years later, in 2001 and after several obstacles and struggles, that the regulation of urban policy was added under the title of "City Statute", bringing a novelty to the connection between urban and environmental policy, especially in Article 2 [17]. Nevertheless, although expressly a city statute, it emphasizes the "right to sustainable cities"; indeed, it is verifiable that there was an ignorance on the part of the enforcers regarding the promotion of strictly urbanized cities devoid of environmental elements [24]. It happens that the current global scenario has undergone an unsustainable change, especially when related to the themes of socio-spatial segregation [2,9].

Thinking and rethinking strategies about urban policies has become necessary, indispensable and, above all, essential to understanding the dynamics of the social functions of the city that are associated with the RTTC approach [28]. It is from this perspective that the social functions of the city, when translated into the functionalities that are executed by the municipal organization and urban policy, gain the new appearance of promoting urban sustainability [19]. In this context, urban sustainability has a specific functionality;

it promotes improvements in urban spaces [29] by transforming them and renews them in order to achieve a new model for sustainable cities [24]. Therefore, the role played by those social functions pursued by Brazilian urban policy in order to promote the basic guidelines for urban sustainability and simultaneously strive for the effectiveness of the RTTC is evident [30,31]. In addition, it is important to emphasize that urban sustainability is not only a normative expression that integrates the Brazilian legal system, but is also one of the 17 sustainable development goals (SDGs) that make up the United Nations' 2030 Agenda. This is especially characterized in SDG 11 [10], which ratifies the relevance of the theme with regard to promoting technical–scientific discussions and reflections on the subject [32].

## 4. Conclusions

Although the urban policy enshrined by the Brazilian Federal Constitution of 1988 is a theoretical and practical apparatus for the realization of the RTTC, it is not to be confused with its objective, which is the right to access cities. Such access occurs as an unfolding of the social functions of a city that, in this text, are classified as the interrelationship between urban policy and the RTTC. Given this, it is also emphasized that the function of this research overview is the construction of a set of ideas that can subsidize and contribute to the theoretical discussions about the RTTC and can set an objective to be achieved by Brazilian urban policy.

In summary, the conclusions here provide two major insights. The first consists of asserting the relevance of urban policy agendas that strive for the effectiveness of the RTTC, and understanding it as the minimum that is required in order to live with dignity and access it in a way that maximizes the benefits that come from urban space. The second insight is the conclusion that there is no intention to exhaust the discussions, reflections and strategic manifestations on this theme. It is necessary to continue these discussions presented here and continue to consider the socio-spatial, socio-environmental and legal context of cities. Therefore, it is necessary that new research be carried out to investigate the existing interface between urban policy and its functionalities in cities, as well as boost access to urbanized and renewed land and thus directly validate the right to sustainable cities.

Finally, this entry proposes that researchers strive for continuity in their discussions and analyze the existing gaps in order to identify new perspectives and issues for examination and debate them using an interdisciplinary technical–scientific approach. It is suggested that further research be directed towards the scope of urban sustainability and its association with various dynamics, such as environmental, social and governmental, as well as the SDGs and the particularities of the New Urban Agenda of the UN—(NUA).

**Author Contributions:** Conceptualization, F.T.D. and M.E.L.; Model, F.T.D., M.E.L., P.C., R.S.B. and J.B.S.O.d.A.G.; software, F.T.D., M.E.L., P.C., R.S.B. and J.B.S.O.d.A.G.; validation, F.T.D.; formal analysis, F.T.D., M.E.L., P.C., R.S.B. and J.B.S.O.d.A.G.; investigation, F.T.D., M.E.L., P.C., R.S.B. and J.B.S.O.d.A.G.; resources, F.T.D., M.E.L.; P.C., R.S.B. and J.B.S.O.d.A.G.; data curation, F.T.D., M.E.L.; P.C., R.S.B. and J.B.S.O.d.A.G.; writing—original draft preparation, F.T.D., M.E.L., P.C., R.S.B. and J.B.S.O.d.A.G.; writing—review and editing, F.T.D., M.E.L., P.C., R.S.B. and J.B.S.O.d.A.G.; visualization, F.T.D., M.E.L., P.C., R.S.B. and J.B.S.O.d.A.G.; supervision, F.T.D.; project administration, F.T.D. All authors have read and agreed to the published version of the manuscript.

**Funding:** This research received no external funding.

**Informed Consent Statement:** Not applicable.

**Data Availability Statement:** All research data are bibliographic and are available in the references of this paper.

**Acknowledgments:** This study was conducted by Rede Colaborativa de Pesquisa em Sustentabilidade Urbana & Direito à Cidade/Núcleo de Direito à Cidade do Observatório UniFG do Semiárido Nordestino, Programa de Pós-graduação em Desenvolvimento Social da Universidade Estadual de Montes Claros/MG, Centre for Sustainable Development (Greens). Special Acknowledgments to Conselho Nacional de Desenvolvimento Científico e Tecnológico—CNPq; Fundação de Amparo à Pesquisa do Estado de Minas Gerais—FAPEMIG and Universidade Estadual de Montes Claros–MG.

**Conflicts of Interest:** The authors declare no conflict of interest. The funders had no role in the design of the study; in the collection, analyses, or interpretation of data; in the writing of the manuscript; or in the decision to publish the results.

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
