# Peer review of "Brazilian Urban Policy: Sustainability as a Driving Force"

_encyclopedia, doi:10.3390/encyclopedia3020044_

Round 1
Reviewer 1 Report
Thanks to the authors for their entry submission. ‘Brazilian Urban Policy: Sustainability as A Driving Force’ is an interesting topic. However, this entry has major issues as mentioned below:
Abstract needs to be improved with rewording of the main research aim and key/specific findings.
I would like the authors to clarify the aim of the study at the end of the Introduction section.
A stronger case and rationale for the study needs to be developed.
In the introduction section, it is stated that Social Functions (SF) reflect on the idea of sustainable cities, their linkage and relationship need to be set out for further clarity.
Section 1.1 discusses Delimiting Urban Policy. The other themes related to the research such as sustainability, sustainable cities and Right to the City are not critically reviewed appropriately. For example, sustainability needs more attention in the context of urban policy.
Table captions need to be in English, for example, page 3.
Methodology needs some more specific details for clarity with appropriate justifications.
In Section 2, research approach and methodology are presented. However, there are gaps in terms of clarifying the actual method/s used. Which key terms were used, which databases were used, and all other practical details need to be specified for clarity of the reader.
Overall, the literature review is missing deeper analysis and critique in line with the aims of the study which has also impacted conclusions to be drawn.
The arguments need coherence in the later part of the entry and clear focus is required on the main themes of the study.
Conclusions also need to be revised based on the analysis with some implications and future research direction.
This entry has some major issues with writing style and English language. It has made it difficult to follow as a reader. Therefore, this needs to be reworked/re-written throughout. Also, proofread the entry to address all the English language related errors and issues.
Author Response
Thanks to the authors for their entry submission. ‘Brazilian Urban Policy: Sustainability as A Driving Force’ is an interesting topic. However, this entry has major issues as mentioned below:
- Abstract needs to be improved with rewording of the main research aim and key/specific findings.
Answer: Dear Reviewer, we cordially thank you for the suggestion, the abstract has been entirely redone, however, considering the nuances of the type of study, that is, Entry Paper. It can be seen in the manuscript.
- I would like the authors to clarify the aim of the study at the end of the Introduction section.
Answer: Dear Reviewer, as described at the end of the introduction, the objective established specifically for this entry is in fact to promote reflections on urban sustainability through an interdisciplinary dynamic and linking aspects of Brazilian Urban Policy with other global guidelines, such as the New Urban Agenda and the 2030 Agenda, emphasizing analyzes on the connection of urban policies and Social Functions (SF) reflecting on the idea of sustainable cities.
- A stronger case and rationale for the study needs to be developed.
Answer: Dear reviewer, thank you for pointing this out. However, since this is an introductory article whose main purpose is to point out a history of a topic, and even discoveries, there is not necessarily a discussion around a case, or even justification for the study, the study in itself is already justified for being a historical and consolidated analysis.
- In the introduction section, it is stated that Social Functions (SF) reflect on the idea of sustainable cities, their linkage and relationship need to be set out for further clarity.
Answer: This connection is demonstrated throughout the text.
- Section 1.1 discusses Delimiting Urban Policy. The other themes related to the research such as Sustainability, sustainable cities and Right to the City are not critically reviewed appropriately. For example, sustainability needs more attention in the context of urban policy.
Answer: Dear Reviewer, regarding item 1.1, it is in fact a brief delimitation of the Urban Policy, above all, articulating it from a Brazilian perspective, which brings some differences. Regarding the other aspects inherent to the Urban Policy, such as sustainable cities and the right to the city, each one of them received specific reports within the text for a better elucidation, in addition, later the discussion about these terms are revisited and unified.
- Table captions need to be in English, for example, page 3.
Answer: Thanks for the note. We make the adjustments as mentioned.
- Methodology needs some more specific details for clarity with appropriate justifications.
Answer: Dear Reviewer, we appreciate the suggestion, however, it is an Entry Paper, for which there is not necessarily a methodology to be used. In this study itself, we used a Model that was resized for Topic 1. We apologize for the confusion.
- In Section 2, research approach and methodology are presented. However, there are gaps in terms of clarifying the actual method/s used. Which key terms were used, which databases were used, and all other practical details need to be specified for clarity of the reader.
Answer: Dear Reviewer, thank you for pointing this out. However, it is an Entry Paper, that is, there is not necessarily a methodology to be used. In this study itself, we used a Model that was resized for Topic 1. We apologize for the confusion.
- Overall, the literature review is missing deeper analysis and critique in line with the aims of the study which has also impacted conclusions to be drawn.
Answer: This entry was built on a dynamic between global aspects of literature and Brazilian legislation that integrates various aspects and concepts that are interesting to Urban Policy, such as: right to the city, sustainable cities and the promotion of urban sustainability. Thus, the propositions established by this manuscript do not correspond to either a systematic review or a narrative review of the literature.
That said, it is necessary to emphasize that to build the entry we used a consistent integration between the main authors/theories that discuss urban policy in the Brazilian scenario. Keep in mind that the Brazilian model is used as a parameter. Therefore, there is no pretense here in a deepening as suggested by the reviewer, in the sense that new studies may in fact suggest new paths and methods (which is not the focus of this entry).
- Conclusions also need to be revised based on the analysis with some implications and future research direction.
Answer: We appreciate your suggestion. We made some adjustments, although the manuscript is an input article, it is pertinent to point out new future studies.
- This entry has some major issues with writing style and English language. It has made it difficult to follow as a reader. Therefore, this needs to be reworked/re-written throughout. Also, proofread the entry to address all the English language related errors and issues.
Answer: Dear reviewer, thank you very much for your notes on the formal errors related to the “Entry Paper” type of manuscript, these have been duly redirected and reorganized. Regarding the vices of the English language, the manuscript was translated by a native, and this one, double-checked the entire manuscript. Thanks again for your time and willingness to review.
Reviewer 2 Report
The manuscript is a scholar study focusing on Brazilian Urban Policy, thus the subject of the manuscript is currently an interesting issue. The title is appropriate for the content of the manuscript. The abstract is concise and accurately summarizes the essential information of the paper. The manuscript complies with journal’s guidance in general, but could be further improved if undergone to some revisions.
More specifically, the literature review is feeble and the manuscript references need also to be enhanced and eventually supported with references of similar studies globally.
Author Response
The manuscript is a scholar study focusing on Brazilian Urban Policy, thus the subject of the manuscript is currently an interesting issue. The title is appropriate for the content of the manuscript. The abstract is concise and accurately summarizes the essential information of the paper. The manuscript complies with journal’s guidance in general, but could be further improved if undergone to some revisions.
More specifically, the literature review is feeble and the manuscript references need also to be enhanced and eventually supported with references of similar studies globally.
Answer: Regarding the references, these were selected based on their relevance to discuss Brazilian urban policy. There is no way to directly correlate a more global bibliography with specific dynamics such as Brazilian legislation and parameters, it would look like the use of a deductive method, which is not the purpose of this entry. However, I find the reviewer's notes very interesting, and as a suggestion, they will be taken into account for future proposals, building a systematic/integrative literature search in depth and developing strategies to correlate global aspects with local/national ones.
Reviewer 3 Report
The paper proposes a serious and up-to-date research approach to substantiate the idea that sustainability is a driving force in Brazilian urban politics. The purpose declared by the authors is "to promote deliberations on urban sustainability through an interdisciplinary dynamic by considering an analysis based on the connection of urban policies and social functions that reflect the idea of ​​sustainable cities". The framework for analysis is mainly Brazilian constitutional provisions and a special law dedicated to cities, the "Cities Law". Starting from this, the authors develop a simple research structure that contains several interactions with studies that have the same objective or a partially complementary objective. Although the authors claim that the research strategy has a dual aspect - exploratory and descriptive - in the few pages of the work we do not find the details necessary to support the announced research methodology. In Table 1 (without title in English) we note 7-8 "main authors" whose works, in the majority, are from the period 2001-2016. Therefore, the so-called "model" is described in 16 lines from which we cannot retain descriptions consistent with the announced research objectives. The same situation can be found in the other subchapters. In conclusion, I consider that the content and methodology presented in the paper do not support the proposed research objectives. The authors use a limited and out-of-date bibliography. There is a lack of analyzes consistent with global approaches to sustainability. The conclusions are irrelevant for the theme of the work, announcing, in most of them, the need for future studies. In this context, I consider that the work must be redone in terms of content, broadening the analysis and integrating the specific national issue in a more obvious and edifying international context.Author Response
- The paper proposes a serious and up-to-date research approach to substantiate the idea that sustainability is a driving force in Brazilian urban politics. The purpose declared by the authors is "to promote deliberations on urban sustainability through an interdisciplinary dynamic by considering an analysis based on the connection of urban policies and social functions that reflect the idea of ​​sustainable cities". The framework for analysis is mainly Brazilian constitutional provisions and a special law dedicated to cities, the "Cities Law". Starting from this, the authors develop a simple research structure that contains several interactions with studies that have the same objective or a partially complementary objective.
Answer: We appreciate the comments, and add that our work is fully connected with the aforementioned legislation, however, it is also attentive to the dynamics of consolidated concepts from global, however, national literatures. Considering above all the type of manuscript as input.
- Although the authors claim that the research strategy has a dual aspect - exploratory and descriptive - in the few pages of the work we do not find the details necessary to support the announced research methodology. In Table 1 (without title in English) we note 7-8 "main authors" whose works, in the majority, are from the period 2001-2016. Therefore, the so-called "model" is described in 16 lines from which we cannot retain descriptions consistent with the announced research objectives. The same situation can be found in the other subchapters.
In conclusion, I consider that the content and methodology presented in the paper do not support the proposed research objectives. The authors use a limited and out-of-date bibliography. There is a lack of analyzes consistent with global approaches to sustainability. The conclusions are irrelevant for the theme of the work, announcing, in most of them, the need for future studies. In this context, I consider that the work must be redone in terms of content, broadening the analysis and integrating the specific national issue in a more obvious and edifying international context.
Answer: Thank you for pointing this out, and we apologize for a formal error, mainly in the repositioning of some aspects and ideas, which the reviewer mentions as “methodology”. The work itself is an input article, and therefore does not have a methodological structure, but a model that, in itself, is not referentially linked to a methodology, but to Chapter 1. Therefore, we repositioned these aspects and delete others. About the references, although they are from previous periods, they are extremely relevant considering the Brazilian dynamics, especially from the perspective of interdisciplinary studies under the legal scope. About connections and disconnections with methodology, I say again, this is not an article that has a methodological structure, but composed of theme, definition and consolidated approach. The up-to-dateness of the references could be a preponderant factor if it were a review article, or even a field/integrative work, which is not the case in point.
Round 2
Reviewer 1 Report
Thanks to the authors for revising the entry and resubmission. I can see the improvements made. I support the publication. However, I still think that English writing needs more improvement. Some of the sentences are unclear and some of the words should be changed for clarity. For example, it should be 'themes' in Table 1 instead of ‘thematic’. Lines 55-59 are not clear. These are just a couple of examples to illustrate. The entry needs to be read properly for proofreading to address any language errors and arguments need to be made very clear.
Author Response
We appreciate the support of the evaluator, and I confirm the requested adjustments on English that were made by a native speaker, specifically from the UK. Also, regarding the specific sentence, from line 55-59, we confirm the adjustments!
Reviewer 3 Report
I reread the manuscript and the changes made.
I think it meets the criteria to be published.
Author Response
We appreciate the contributions and the indication, all the previous and current adjustments have certainly improved our work!
Round 3
Reviewer 1 Report
Thanks to the reviewers for addressing the comments.